# Weight loss during follow-up in patients with acute heart failure: From the KCHF registry

Yuta Seko[1], Takao Kato[1]*, Takeshi Morimoto[2], Hidenori Yaku[1], Yasutaka Inuzuka[3], Yodo Tamaki[4], Neiko Ozasa[1], Masayuki Shiba[1], Erika Yamamoto[1], Yusuke Yoshikawa[1], Takeshi Kitai[5], Yugo Yamashita[1], Moritake Iguchi[6], Kazuya Nagao[7], Yuichi Kawase[8], Takashi Morinaga[9], Mamoru Toyofuku[10], Yutaka Furukawa[11], Kenji Ando[9], Kazushige Kadota[8], Yukihito Sato[12], Koichiro Kuwahara[13], Takeshi Kimura[1]

1 Department of Cardiovascular Medicine, Kyoto University Graduate School of Medicine, Kyoto, Japan, 2 Clinical Epidemiology, Hyogo College of Medicine, Nishinomiya, Japan, 3 Cardiovascular Medicine, Shiga General Hospital, Moriyama, Japan, 4 Division of Cardiology, Tenri Hospital, Tenri, Japan, 5 Department of Cardiovascular Medicine, National Cerebral and Cardiovascular Center, Suita, Japan, 6 Department of Cardiology, National Hospital Organization Kyoto Medical Center, Kyoto, Japan, 7 Department of Cardiology, Osaka Red Cross Hospital, Osaka, Japan, 8 Department of Cardiology, Kurashiki Central Hospital, Kurashiki, Japan, 9 Department of Cardiology, Kokura Memorial Hospital, Kitakyushu, Japan, 10 Department of Cardiology, Japanese Red Cross Wakayama Medical Center, Wakayama, Japan, 11 Department of Cardiovascular Medicine, Kobe City Medical Center General Hospital, Kobe, Japan, 12 Department of Cardiology, Hyogo Prefectural Amagasaki General Medical Center, Amagasaki, Japan, 13 Department of Cardiovascular Medicine, Shinshu University Graduate School of Medicine, Matsumoto, Japan

* tkato75@kuhp.kyoto-u.ac.jp

**Data Availability Statement:** All relevant data are within the paper and its Supporting information files.

## Abstract

### Backgrounds

The prognostic implication of weight loss after discharge from acute heart failure (AHF) remains unclear. We sought to investigate the association of weight loss between discharge and 6-month visit with subsequent clinical outcomes in patients with AHF.

### Methods

We analyzed 686 patients with AHF in the prospective longitudinal follow-up study derived from the Kyoto Congestive Heart Failure registry, and divided them into 2 groups based on the weight loss at 6-month index visit. We defined the weight loss as ≥ 5% decrease in body weight from discharge to 6-month index visit.

### Results

There were 90 patients (13.1%) with a weight loss at 6-month visit. Patients in the weight loss group compared with those in the no weight loss group had higher body weight at discharge and lower body weight at 6-mont visit. Patients in the weight loss group had a lower systolic blood pressure, higher brain-type natriuretic peptide, lower serum albumin, lower hemoglobin, higher prevalence of heart failure with reduced ejection fraction at 6-month visit, and a lower prescription rate of inhibitors of renin-angiotensin system than those in the no weight loss group. The cumulative 6-month incidence of all-cause death was significantly higher in the weight loss group than in the no weight loss group (14.2% and 4.3%, log-rank

**Funding:** This study was supported by grant 18059186 from the Japan Agency for Medical Research and Development (Drs T. Kato, Kuwahara, and Ozasa).

**Competing interests:** The authors have declared that no competing interests exist.

P<0.001). The excess adjusted risk of the weight loss group relative to the no weight loss group remained significant for all-cause death (HR 2.39, 95%CI 1.01–5.65, P = 0.048).

## Conclusion

Body weight loss of $\geq$5% at 6-month visit after discharge was associated with subsequent all-cause death in patients with AHF.

## Introduction

The clinical and prognostic implications of cardiac cachexia in patients with chronic heart failure (CHF) have been the subject of many reports, concluding that weight loss as cardiac cachexia is a significant risk for mortality and should be targeted with therapeutic and preventive interventions [1–4]. It is easy to record changes in body weight, but it is frequently difficult to interpret the changes in patients with acute HF (AHF), because the body weight change may be influenced by the congestive status of each patient which is often unstable in AHF. In this context, there is a paucity of data on the incidence and effect of weight loss in patients with AHF.

When we follow the patients with AHF, we usually check the body weight at a follow-up visit and compare it to that at discharge. Thus, we investigated the incidence of weight loss after discharge and the association between weight loss and clinical outcomes in patients discharged for AHF using data from a large Japanese registry.

## Materials and methods

### Study design, setting, and population

The KCHF (Kyoto Congestive Heart Failure) registry is a physician-initiated, prospective, observational, multicenter cohort study enrolling consecutive patients who were admitted to the hospital due to AHF for the first time between October 2014 and March 2016. In parallel with the main KCHF study, we designed a prospective, longitudinal study enrolling a subgroup of patients from the KCHF study in which the selected patients were to have a 6-month visit with an allowance of 1-month (Fig 1A) [5]. At the follow-up visit, we collected the data for physical findings, echocardiography, laboratory data and medications at 6-month after enrollment. Time zero for the clinical follow-up in the present study was the day of the 6-month follow-up visit, and considered it as the index day. Clinical follow-up was censored at 210 days after the 6-month visit (Fig 1B). Exclusion criteria for the prospective longitudinal follow-up study were as follows: no written informed consent (N = 238), patient with an age of <20 years (N = 1), fever or infectious diseases at admission (N = 297), acute coronary syndrome at admission (N = 157), end-stage renal failure (N = 218), severe comorbidity limiting the life expectancy within one year assessed by attending physicians at each participating center, such as end-stage cancer, severe cognitive dysfunction, and end-stage liver dysfunction (N = 112), ineligible for follow up (unable to visit each participating hospital for various reasons) (N = 1516). After excluding 271 patients who died during index hospitalization and 2539 patients who did not meet the pre-specified criteria of follow-up, 1246 patients were enrolled in the prospective longitudinal follow-up study. We excluded 23 patients who died within 6 months after initial hospitalization, 14 patients with lost to follow-up during 6 months after enrollment, 250 patients without data for 6-month visit, 273 patients with missing body weight data at discharge and/or at 6-month visit (Fig 1A and S1 Table). Therefore, the current study

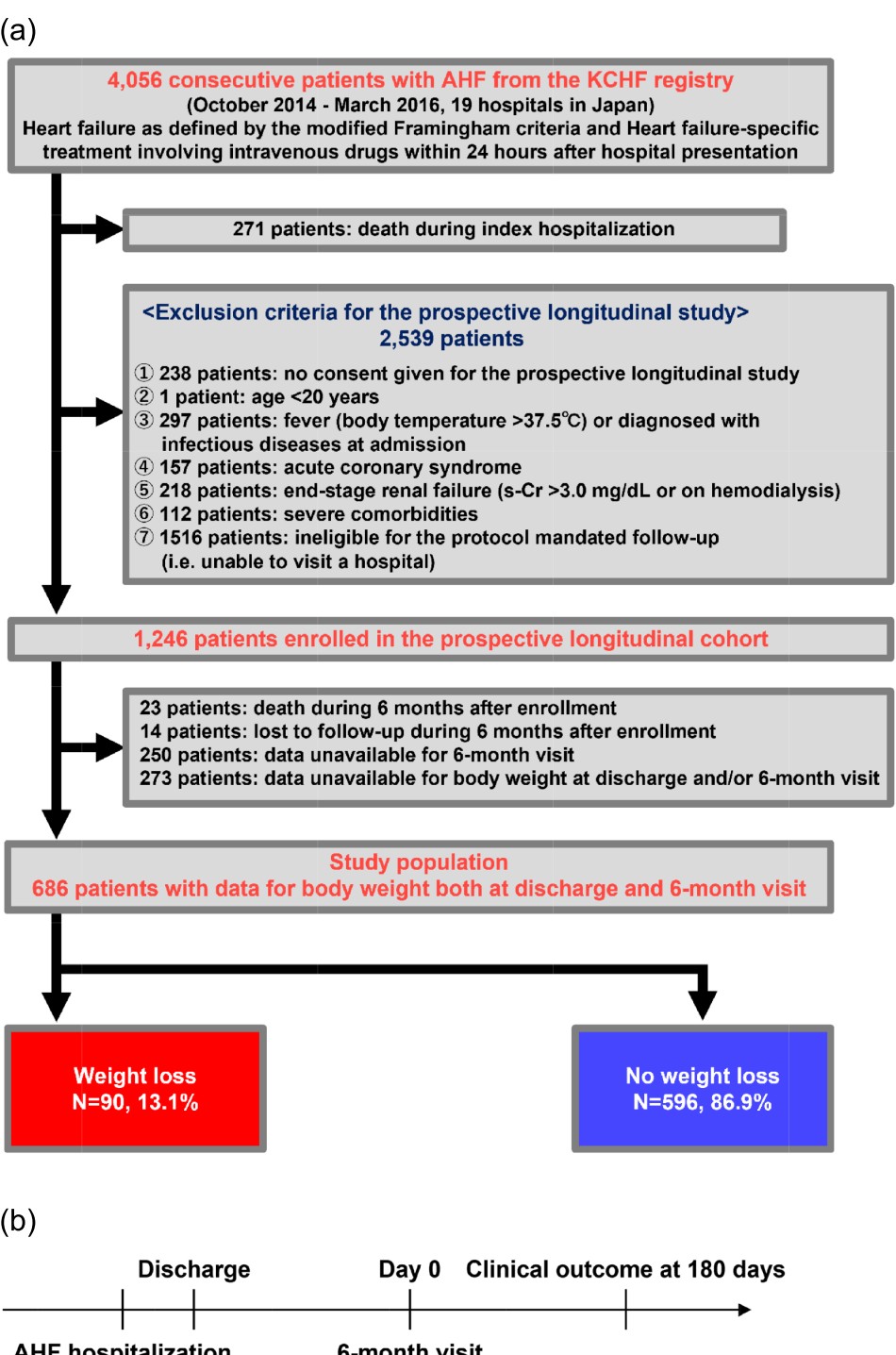

**Fig 1. Study patients flow.** (A) Study patients flow, and (B) Time course of the study. AHF, acute heart failure; KCHF, Kyoto Congestive Heart Failure; s-Cr, serum creatinine.

population consisted of 686 patients with available data for body weight both at discharge and at 6-month visit. We stratified the patients into 2 groups as follows: the weight loss and no weight loss group.

## Ethics

The present investigation conforms to the principles outlined in the Declaration of Helsinki. The study protocol was approved by the ethical committees at the Kyoto University Hospital (local identifier: E2311), as well as at each participating hospital. Written informed consent was obtained from the patients enrolled in the longitudinal prospective cohort study. We made identifiable patient data anonymous before the analysis.

## Definitions

The changes (delta, Δ) for body weight and other variables were calculated according to the following equation: (the value at 6-month visit)—(the value at discharge). We defined the weight loss as $\geq$ 5% decrease in body weight in the main analysis [4,6]. In the sensitivity analysis, we defined the body weight changes as follows; weight loss = $\geq$ 5% decrease in body weight, no weight change [reference] = -5% <body weight change < 5% and weight gain = $\geq$ 5% increase in body weight (S1 Fig). We additionally classified into three groups as follows: $\geq$10% decrease in body weight, -10%< body weight change $\leq$ -5% and no weight loss [reference] (-5%< body weight change) (S2 Fig). The detailed definitions of baseline patient characteristics were previously described [5,7]. Anemia was defined using the World Health Organization criteria (hemoglobin <12.0 g/dL in women and <13.0 g/dL in men). Chronic kidney disease was defined as estimated glomerular filtration rate (eGFR) <60 mL/min/1.73 $m^2$. End-stage renal disease was defined as eGFR <30 mL/min/1.73 $m^2$ based on the chronic kidney disease grades [7]. Geriatric nutritional risk index (GNRI) was calculated as follows: [14.89 x serum albumin levels (g/dL)] + [41.7 x (BW/ ideal BW)] and was defined <92 as moderate or major risk of malnutrition.

## Outcomes

The primary outcome measure for the present analysis was all-cause death after 6-month visit. Secondary outcome measure was HF hospitalization. HF hospitalization was defined as hospitalization due to worsening of HF requiring intravenous drug therapy [7].

## Statistical analysis

Categorical variables were presented as numbers with percentages and compared using the chi-square test. Continuous variables were expressed as mean with standard deviation (SD) or median with interquartile range (IQR) and compared using the Student's t test or Wilcoxon rank sum test based on their distribution.

The cumulative incidences of the clinical events after the 6-month visit were estimated using the Kaplan Meier (KM) method with the between-groups difference assessed by the log-rank test. The cumulative incidence of HF hospitalization after discharge was estimated using the Gray method, accounting for the competing risk of all-cause death. Multivariable Cox proportional hazards model was developed to estimate the adjusted risk of weight loss relative to no weight loss for the primary outcome measure. To account for the competing risk of all-cause death, the risk of HF hospitalization was described using the Fine-Gray subdistribution hazard model. We included the following 11 clinically relevant risk-adjusting variables into the model: age $\geq$80 years, sex, malignancy, HF with reduced ejection fraction (HFrEF: left

ventricular ejection fraction [LVEF]<40%), body mass index (BMI) <20 kg/m$^2$ at 6-month visit, estimated glomerular filtration rate (eGFR) <30 ml/min/1.73 m$^2$ at 6-month visit, albumin <3.0 g/dL at 6-month visit, anemia at 6-month visit and medications at 6-month visit (angiotensin converting enzyme inhibitors [ACEIs] or angiotensin II receptor blockers [ARBs], β-blockers, and mineralocorticoid receptor antagonists [MRAs]). We selected these variables according to their clinical relevance to the clinical outcomes and based on the previous studies [8,9]. The continuous variables were dichotomized by the clinically meaningful reference values or median values. We also evaluated the interactions between the subgroup factors such as age, etiology of HF, LVEF, BMI, albumin, anemia and GNRI at 6-month visit and the effects of weight loss relative to no weight loss on all-cause death after 6-month visit. The results were expressed as hazard ratios (HRs) and 95% confidence intervals (CIs). All statistical analyses were conducted by 2 physicians (Y.S. and T.K.) and a statistician (T.M.) using JMP Pro 15 (SAS Institute Inc., Cary, NC, USA) and EZR [10]. All the reported P values were two tailed, and the level of statistical significance was set at P <0.05.

## Results

### Baseline characteristics and medications

There were 90 patients (13.1%) with a weight loss at 6-month visit (Fig 1). The mean body weight was 55.5 ± 14.6 kg at discharge and 56.5 ± 14.8 kg at 6-month visit. Patients in the weight loss group had higher body weight at discharge and lower body weight at 6-mont visit. Patients in the weight loss group had a lower systolic blood pressure, lower serum albumin, lower hemoglobin, lower GNRI with a less increase of GNRI during 6 months, and higher prevalence of HF with reduced EF at 6-month visit (Table 1). Characteristics of the patients at discharge shown in S2 Table.

### Clinical outcomes after 6-month visit

Final follow-up after the 6-month visit was completed in 85.2% of patients.

The cumulative 6-month incidence of the primary outcome measure was 14.2% in the weight loss group, and 4.3% in the no weight loss group (P<0.001) (Fig 2A). The excess adjusted risk of the weight loss group relative to the no weight loss group remained significant for the primary outcome measure (HR 2.39, 95%CI 1.01–5.65, P = 0.048) (Table 2). The cumulative 6-month incidence of HF hospitalization was 7.6% in the weight loss group, and 11.1% in the no weight loss group (P = 0.21) (Fig 2B). There was no significant excess adjusted risk of the weight loss group relative to the no weight loss group for HF hospitalization (HR 0.74, 95% CI, 0.29–1.92, P = 0.54) (Table 2).

### Sensitivity analysis

When we divided the study population into 3 groups (the weight loss group, N = 90, the no weight change group, N = 402, and the weight gain group, N = 194) (S3 Table), the cumulative 6-month incidence of the primary outcome measure was 14.2% in the weight loss group, 4.6% in the no weight change group, and 3.9% in the weight gain group (P<0.001) (S3 Fig). There was a numerically excess risk of the weight loss group relative to the no change group for all-cause death (HR 2.29, 95%CI 0.94–5.58, P = 0.07), while there was no excess adjusted risk of the weight gain group relative to the no change group for all-cause death (HR 0.83, 95%CI, 0.29–2.41, P = 0.74) (S3 Fig).

When we divided the study population into another 3 groups (≥10% decrease in body weight, N = 36, -10%< body weight change ≤ -5%, N = 54 and no weight loss [-5%< body

**Table 1. Characteristics and transthoracic echocardiography results at discharge and at 6-month visit.**

| | Total (N = 686) | Weight loss (N = 90) | No weight loss (N = 596) | P value | Evaluable N |
|---|---|---|---|---|---|
| **Clinical Characteristic** | | | | | |
| Age, years | 78 (70–84) | 79 (72–84) | 77 (69–84) | 0.27 | 686 |
| Age≥80 years* | 291 (42.4) | 40 (44.4) | 251 (42.1) | 0.68 | 686 |
| Men* | 405 (59.0) | 50 (55.6) | 355 (59.6) | 0.47 | 686 |
| Body weight at discharge, kg | 55.5 ± 14.6 | 59.3 ± 19.4 | 54.9 ± 13.7 | 0.009 | 686 |
| Body weight at 6-month visit, kg | 56.5 ± 14.8 | 53.3 ± 16.4 | 57.0 ± 14.5 | 0.03 | 686 |
| ⊿Body weight | 1.0 ± 4.8 | -6.0 ± 4.6 | 2.1 ± 3.8 | <0.001 | 686 |
| BMI at discharge | 22.3 ± 4.7 | 24.0 ± 6.9 | 22.0 ± 4.3 | <0.001 | 676 |
| BMI at 6-month visit | 22.7 ± 4.7 | 21.6 ± 5.7 | 22.8 ± 4.5 | 0.02 | 676 |
| BMI<20 at 6-month visit* | 201 (29.7) | 39 (44.8) | 162 (27.5) | 0.001 | 676 |
| **Etiology** | | | | | |
| Ischemic | 199 (29.0) | 26 (28.9) | 173 (29.0) | 0.98 | 686 |
| **Medical history** | | | | | |
| Hypertension | 507 (73.9) | 71 (78.9) | 436 (73.2) | 0.25 | 686 |
| Diabetes | 258 (37.6) | 36 (40.0) | 222 (37.2) | 0.62 | 686 |
| Dyslipidemia | 294 (42.9) | 41 (45.6) | 253 (42.5) | 0.58 | 686 |
| Atrial fibrillation or flutter | 376 (54.8) | 49 (54.4) | 327 (54.9) | 0.94 | 686 |
| Previous myocardial infarction | 171 (24.9) | 24 (26.7) | 147 (24.7) | 0.68 | 686 |
| Previous stroke | 113 (16.5) | 16 (17.8) | 97 (16.3) | 0.72 | 686 |
| Chronic kidney disease | 304 (44.3) | 48 (53.3) | 256 (43.0) | 0.06 | 686 |
| Chronic lung disease | 88 (12.8) | 11 (12.2) | 77 (12.9) | 0.85 | 686 |
| Malignancy* | 102 (14.9) | 19 (21.1) | 83 (13.9) | 0.07 | 686 |
| Cognitive dysfunction | 73 (10.6) | 10 (11.1) | 63 (10.6) | 0.88 | 686 |
| **Vital signs at 6-month visit** | | | | | |
| Heart rate, bpm | 74.7 ± 13.8 | 76.2 ± 15.3 | 74.4 ± 13.5 | 0.27 | 653 |
| Systolic BP, mmHg | 121.2 ± 21.6 | 114.7 ± 26.7 | 122.2 ± 20.5 | 0.002 | 661 |
| Diastolic BP, mmHg | 67.7 ± 13.5 | 65.9 ± 14.9 | 68.0 ± 13.2 | 0.18 | 660 |
| NYHA class III or IV | 32 (7.0) | 6 (10.7) | 26 (6.5) | 0.24 | 458 |
| **Test at 6-month visit** | | | | | |
| LVEF, % | 50.6 ± 16.1 | 49.6 ± 17.8 | 50.8 ± 15.8 | 0.52 | 630 |
| HFrEF (LVEF<40%)* | 166 (26.3) | 29 (35.8) | 137 (25.0) | 0.04 | 630 |
| ⊿LVEF, % | 6.1 ± 13.4 | 2.7 ± 12.3 | 6.6 ± 13.5 | 0.02 | 627 |
| BNP, pg/ml | 181 (78–382) | 241 (103–480) | 173 (73–365) | 0.06 | 527 |
| ⊿BNP, pg/ml | -26 ± 308 | -38 ± 346 | -24 ± 302 | 0.76 | 436 |
| NT-proBNP, pg/ml | 1156 (545–2611) | 1014 (285–2254) | 1181 (559–2629) | 0.43 | 267 |
| Serum creatinine, mg/dl | 1.14 (0.89–1.57) | 1.21 (0.84–1.66) | 1.14 (0.90–1.54) | 0.44 | 673 |
| ⊿creatinine, mg/dl | 0.09 ± 0.41 | 0.04 ± 0.41 | 0.09 ± 0.41 | 0.24 | 668 |
| eGFR, ml/min/1.73m$^2$ | 45.3 ± 20.6 | 43.0 ± 18.9 | 45.6 ± 20.8 | 0.28 | 673 |
| <30 ml/min/1.73m$^{2*}$ | 161 (23.9) | 22 (25.0) | 139 (23.8) | 0.80 | 673 |
| ⊿eGFR, ml/min/1.73m$^2$ | -2.3 ± 12.9 | -1.4 ± 13.0 | -2.4 ± 12.9 | 0.49 | 668 |
| Albumin, g/dl | 3.91 ± 0.49 | 3.75 ± 0.54 | 3.94 ± 0.48 | 0.002 | 628 |
| <3.0 g/dl* | 17 (2.7) | 5 (6.3) | 12 (2,2) | 0.04 | 628 |
| ⊿Albumin, g/dl | 0.40 ± 0.47 | 0.37 ± 0.43 | 0.40 ± 0.48 | 0.65 | 580 |
| Sodium, mEq/l | 139.6 ± 3.3 | 139.3 ± 3.3 | 139.6 ± 3.3 | 0.41 | 669 |
| <135 mEq/l | 42 (6.3) | 7 (8.0) | 35 (6.0) | 0.49 | 669 |
| ⊿Sodium, mEq/l | 0.8 ± 3.5 | 0.3 ± 3.2 | 0.9 ± 3.6 | 0.18 | 661 |

(*Continued*)

**Table 1.** (Continued)

| | Total (N = 686) | Weight loss (N = 90) | No weight loss (N = 596) | P value | Evaluable N |
|---|---|---|---|---|---|
| Hemoglobin, g/dl | 12.0 ± 2.1 | 11.5 ± 2.1 | 12.1 ± 2.2 | 0.01 | 671 |
| Anemia* | 396 (59.0) | 57 (64.0) | 339 (58.2) | 0.30 | 671 |
| ⊿Hemoglobin, g/dl | -0.1 ± 1.9 | -0.3 ± 1.8 | -0.1 ± 1.9 | 0.27 | 655 |
| **Medication at 6-month visit** | | | | | |
| ACEIs or ARBs* | 337 (58.9) | 36 (49.3) | 301 (60.3) | 0.07 | 572 |
| β-blockers* | 440 (76.8) | 59 (79.7) | 381 (76.4) | 0.52 | 573 |
| MRAs* | 266 (46.7) | 40 (54.1) | 226 (45.6) | 0.17 | 570 |
| Diuretics | 482 (84.0) | 66 (89.2) | 416 (83.2) | 0.19 | 574 |
| **Nutritional score at 6-month visit** | | | | | |
| GNRI | 101.4 ± 12.7 | 96.5 ± 14.4 | 102.1 ± 12.2 | <0.001 | 620 |
| <92 | 134 (21.6) | 32 (41.0) | 102 (18.8) | <0.001 | 620 |
| ⊿GNRI | 6.8 ± 8.2 | 1.5 ± 6.9 | 7.5 ± 8.1 | <0.001 | 572 |

Values are number (%), mean ± standard deviation (SD), or median (Interquartile range). P values were calculated using the chi square test for categorical variables, and the Student's t test or Wilcoxon rank sum test for continuous variables. The changes (delta, ∆) were calculated according to the following equation: (the value at 6-month visit)—(the value at discharge).

ACEI, angiotensin-converting enzyme inhibitor; ARB, angiotensin-receptor blocker; BMI, body mass index; BP, blood pressure; BNP, brain-type natriuretic peptide; eGFR, estimated glomerular filtration rate; GNRI, geriatric nutritional risk index; HFrEF, heart failure with reduced ejection fraction; LVEF, left ventricular ejection fraction; MRA, mineralocorticoid receptor antagonist; NT-proBNP, N-terminal pro-brain-type natriuretic peptide; NYHA, New York Heart Association.

* Risk-adjusting variables selected for the Cox proportional hazard model and the Fine-Gray subdistribution hazard model.

weight change], N = 596) (S4 Table), the cumulative 6-month incidence of the primary outcome measure was 23.6% in the ≥10% decrease in body weight group, 7.8% in the -10%< body weight change ≤ -5% group, and 4.3% in the no weight loss group (P<0.001) (S4 Fig). There was excess risk of the ≥10% decrease in body weight group relative to the no weight loss group for all-cause death (HR 3.91, 95%CI 1.33–11.49, P = 0.01), while there was no excess adjusted risk of the -10%< body weight change ≤ -5% group relative to the no weight loss group for all-cause death (HR 1.61, 95%CI, 0.51–5.06, P = 0.42) (S4 Fig).

## Subgroup analysis

In the subgroup analysis, there was no significant interaction between the effect of weight loss relative to no weight loss on the primary outcome measure and the subgroup factors such as age, etiology of HF, LVEF, BMI, albumin, anemia, and GNRI (S5 Fig).

## Discussion

The main findings of the present study are as follows; 1) 13.1% of the patients had ≥ 5% weight loss at 6-month visit after discharge from AHF; 2) The weight loss at 6-month visit was associated with increased risk for subsequent all-cause death, but not for HF hospitalization.

### Proportion of patients with weight loss in patients with AHF and outcomes

In patients with CHF, 11–16% patients had ≥5% weight loss within 1-year [6,11]. There were few studies that investigated the prognostic impact of weight loss after discharge in patients with AHF. Agra-Bermejo et al reported that 21.7% of the 92 patients with AHF had ≥6% weight loss at 6-month after discharge with a mean BMI of 30 in Spain, who were associated with poor outcomes [12]. In our study, 13.1% of the patients had ≥5% weight loss at 6-month

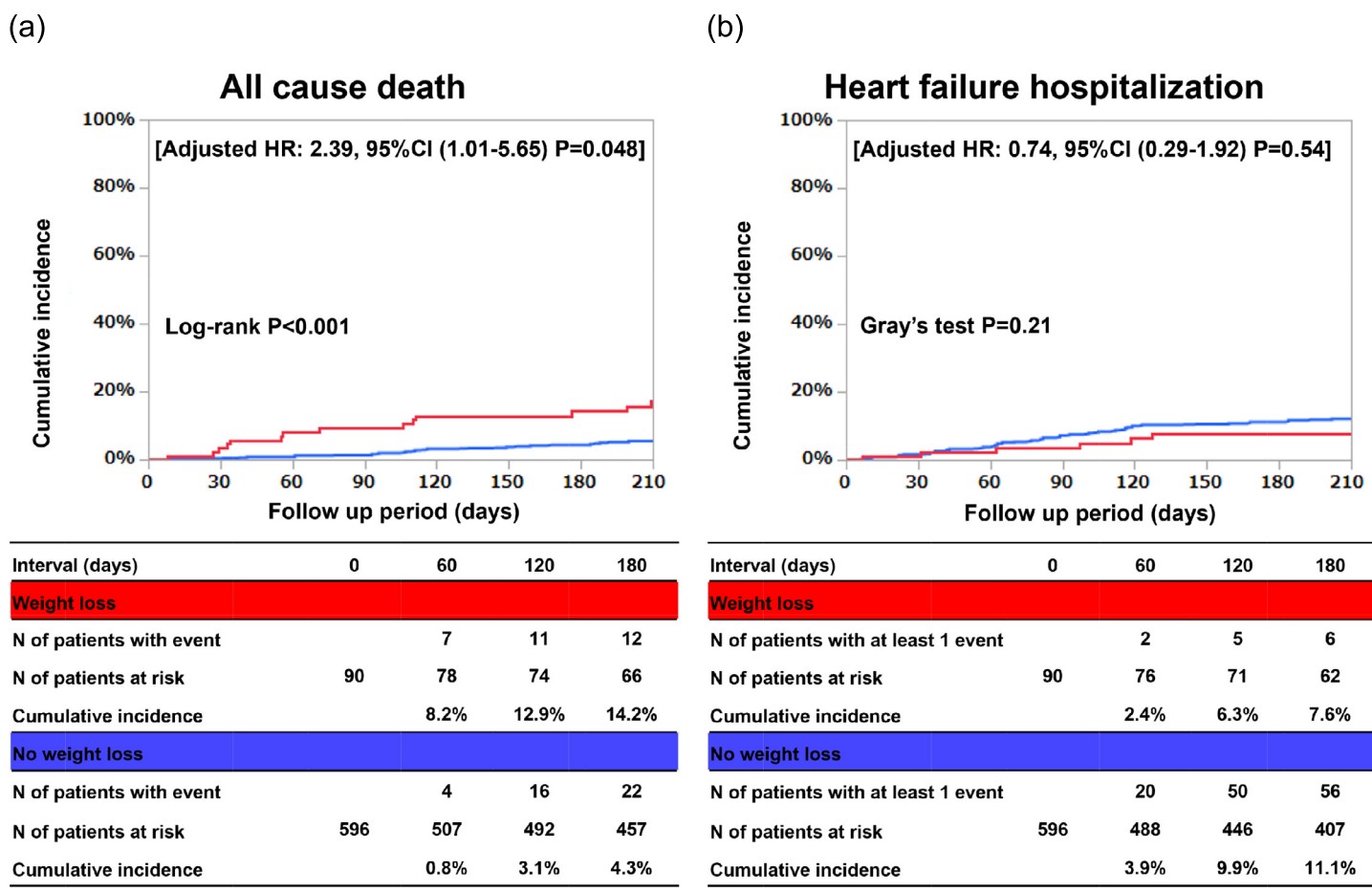

**Fig 2. Kaplan-Meier curves for the primary outcome measure and secondary outcome measure.** (A) The primary outcome measure: All-cause death, and (B) The secondary outcome measure: HF hospitalization. CI, Confidence interval; HF, heart failure; HR, Hazard ratio.

visit with a mean BMI of approximately 22. Despite the differences in patients' backgrounds, a substantial population (from 10% to 20%) of patients with AHF may have ≥5% weight loss during 6-month follow-up and they had a higher risk for adverse events. After adjusting confounders including the BMI, we showed that the weight loss after discharge was associated with increased risk for all-cause death after 6-month visit. It is not clear why the KM curves for all-cause mortality at 30 days differ. There are two suggested reasons for this: 1) the number of

**Table 2. Clinical outcomes.**

| | Weight loss<br>N of patients with event/N of patients at risk<br>(Cumulative 6-month incidence [%]) | No weight loss<br>N of patients with event/N of patients at risk<br>(Cumulative 6-month incidence [%]) | Unadjusted | | Adjusted | |
|---|---|---|---|---|---|---|
| | | | HR (95% CI) | P value | HR (95% CI) | P value |
| All-cause death | 21/90 (14.2%) | 48/596 (4.3%) | 3.38 (1.77–6.45) | <0.001 | 2.39 (1.01–5.65) | 0.048 |
| HF hospitalization | 6/90 (7.6%) | 56/596 (11.1%) | 0.59 (0.25–1.36) | 0.21 | 0.74 (0.29–1.92) | 0.54 |

CI, confidence interval; HF, heart failure; HR, hazard ratio.

patients enrolled in this study is relatively small and the KM curves for the two cohorts are not smoothed; 2) it is possible that some of the patients who lost weight after discharge may have further improved congestion, making it difficult for them to have events immediately after the starting point of 6 months after discharge. Regarding HF hospitalization, we used the Fine-Gray subdistribution hazard model to deal with the competing risk of all-cause death and showed that the weight loss after discharge was not associated with HF hospitalization. It is possible that some of the patients who had weight loss after discharge may have had further improvement in congestion, which may have influenced the results.

## Characteristics in patients with body weight loss in AHF

In our study, patients with weight loss were more likely to have HFrEF, less often received ACEI/ARB, and had higher brain-type natriuretic peptide (BNP) and lower albumin. In GISSI-HF, lower LVEF was associated with ≥5% weight loss within 1 year in patients with CHF [6]. Effective treatment of weight loss including cardiac cachexia has not been reported yet, although several studies targeting weight loss in various diseases including HF are ongoing [3,4]. Clinical trials have demonstrated that ACEIs and beta-blockers promote weight gain with the prevention of cardiac cachexia [1,13]. However, there was no difference in the prescription rate of beta-blockers between the 2 groups in the present analysis. The predictors of weight loss were higher BNP and N-terminal pro-brain-type natriuretic peptide (NT-proBNP) levels at baseline in patients with CHF [6,11,14]. In our study, no difference was observed in NT-proBNP levels at follow-up visit due to the small number of patients with data and large variability, although the increased NT-proBNP concentrations were associated with decreased abdominal fat mass in cardiac cachexia [15,16]. Patients with HF are likely to have reduced appetite and food intake [17]. Reduced food intake and decreased physical activity often causes cardiac cachexia [18]. Low serum albumin is one of the components in diagnostic criteria for cachexia [19]. The increase of GNRI was less in the weight loss group. These data suggest the importance of nutrition management and maintenance of activity during the follow-up for patients with AHF. In addition, there was a trend that patients with low BMI, low albumin levels, or low GNRI at 6-month visit were likely to have more risk when compared to those without, although there was no significant interaction between the effect of weight loss relative to no weight loss on the primary outcome measure and the subgroup factors.

Further research would be warranted to investigate the association between weight loss after discharge and prognosis. Additional information would be needed to develop the strategies for preventing weight loss in patients with HF.

## Limitations

First, body weight and laboratory test at 6-month visit was not available substantially. One of the reasons might be very advanced age of the longitudinal cohort (S1 Table). Second, there might be residual confounding in this study despite of adjusting for 11 relevant variables. Third, patients with acute coronary syndrome and end-stage renal failure were excluded and it may decrease the generalizability of the study due to the selection bias. Fourth, weight loss in this study was assessed at 6-month after discharge, but the optimal time interval for evaluating weight loss remained unclear. Sixth, the lack of physical training is a well-known determinant of both weight loss and poor outcome in HF patients [20], but we have no data about undergoing cardiac rehabilitation or training programs after the index hospitalization.

## Conclusion

Body weight loss of ≥5% at 6-month visit after discharge was associated with subsequent all-cause death in patients with AHF.

## Supporting information

**S1 Fig. Study patients flow in the sensitivity analysis (weight loss, no weight change and weight gain).** AHF, acute heart failure; KCHF, Kyoto Congestive Heart Failure; s-Cr, serum creatinine.
(PDF)

**S2 Fig. Study patients flow in the sensitivity analysis (≥10% decrease in body weight, -10%< body weight change ≤ -5% and no weight loss [-5%< body weight change]).** AHF, acute heart failure; KCHF, Kyoto Congestive Heart Failure; s-Cr, serum creatinine.
(PDF)

**S3 Fig. Kaplan-Meier curves in the sensitivity analysis (weight loss, no weight change and weight gain).** CI, Confidence interval; HR, Hazard ratio.
(PDF)

**S4 Fig. Kaplan-Meier curves in the sensitivity analysis (≥10% decrease in body weight, -10%< body weight change ≤ -5% and no weight loss [-5%< body weight change]).** CI, Confidence interval; HR, Hazard ratio.
(PDF)

**S5 Fig. Subgroup analysis.** BMI, body mass index; CI, confidence interval; GNRI, geriatric nutritional risk index; HR, hazard ratio; LVEF, left ventricular ejection fraction.
(PDF)

**S1 Table. Baseline characteristics of the patients available with body weight data and those unavailable with body weight data at discharge.** Values are number (%), mean ± standard deviation (SD), or median (interquartile range). P values were calculated using the chi square test for categorical variables, and the Student's t test or Wilcoxon rank sum test for continuous variables. ACEI, angiotensin-converting enzyme inhibitor; ARB, angiotensin-receptor blocker; BMI, body mass index; MRA, mineralocorticoid receptor antagonist.
(PDF)

**S2 Table. Baseline characteristics at discharge.** Values are number (%), mean ± standard deviation (SD), or median (interquartile range). P values were calculated using the chi square test for categorical variables, and the Student's t test or Wilcoxon rank sum test for continuous variables. ACEI, angiotensin-converting enzyme inhibitor; ARB, angiotensin-receptor blocker; BMI, body mass index; BP, blood pressure; BNP, brain-type natriuretic peptide; eGFR, estimated glomerular filtration rate; HFrEF, heart failure with reduced ejection fraction; LVEF, left ventricular ejection fraction; MRA, mineralocorticoid receptor antagonist; NT-proBNP, N-terminal pro-brain-type natriuretic peptide; NYHA, New York Heart Association.
(PDF)

**S3 Table. Baseline characteristics in sensitivity analysis (weight loss, no weight change and weight gain).** Values are number (%), mean ± standard deviation (SD), or median (interquartile range). P values were calculated using the chi square test for categorical variables, and 1-way ANOVA or Kruskal-Wallis test for continuous variables. The changes (delta, Δ) were calculated according to the following equation: (the value at 6-month visit)—(the value at

discharge). ACEI, angiotensin-converting enzyme inhibitor; ARB, angiotensin-receptor blocker; BMI, body mass index; BP, blood pressure; BNP, brain-type natriuretic peptide; eGFR, estimated glomerular filtration rate; GNRI, geriatric nutritional risk index; HFrEF, heart failure with reduced ejection fraction; LVEF, left ventricular ejection fraction; MRA, mineralocorticoid receptor antagonist; NT-proBNP, N-terminal pro-brain-type natriuretic peptide; NYHA, New York Heart Association. * Risk-adjusting variables selected for the Cox proportional hazard models and the Fine-Gray subdistribution hazard model.
(PDF)

**S4 Table. Baseline characteristics in sensitivity analysis (≥10% decrease in body weight, -10%< body weight change ≤ -5% and no weight loss[-5%< body weight change]).** Values are number (%), mean ± standard deviation (SD), or median (interquartile range). P values were calculated using the chi square test for categorical variables, and 1-way ANOVA or Kruskal-Wallis test for continuous variables. The changes (delta, Δ) were calculated according to the following equation: (the value at 6-month visit)—(the value at discharge). ACEI, angiotensin-converting enzyme inhibitor; ARB, angiotensin-receptor blocker; BMI, body mass index; BP, blood pressure; BNP, brain-type natriuretic peptide; eGFR, estimated glomerular filtration rate; GNRI, geriatric nutritional risk index; HFrEF, heart failure with reduced ejection fraction; LVEF, left ventricular ejection fraction; MRA, mineralocorticoid receptor antagonist; NT-proBNP, N-terminal pro-brain-type natriuretic peptide; NYHA, New York Heart Association. * Risk-adjusting variables selected for the Cox proportional hazard models and the Fine-Gray subdistribution hazard model.
(PDF)

## Acknowledgments

The authors thank the members of the KCHF study, the other members of the participating centers.

## Author Contributions

**Conceptualization:** Yuta Seko, Hidenori Yaku, Yasutaka Inuzuka, Yodo Tamaki, Neiko Ozasa, Erika Yamamoto.

**Data curation:** Yuta Seko, Takao Kato, Hidenori Yaku, Yasutaka Inuzuka, Yodo Tamaki, Neiko Ozasa, Masayuki Shiba, Erika Yamamoto, Yusuke Yoshikawa, Takeshi Kitai, Yugo Yamashita, Moritake Iguchi, Kazuya Nagao, Yuichi Kawase, Takashi Morinaga, Mamoru Toyofuku, Yutaka Furukawa, Kenji Ando, Kazushige Kadota, Yukihito Sato, Koichiro Kuwahara.

**Formal analysis:** Yuta Seko, Takeshi Morimoto.

**Investigation:** Yuta Seko, Takao Kato, Hidenori Yaku, Yasutaka Inuzuka, Yodo Tamaki, Neiko Ozasa, Masayuki Shiba, Erika Yamamoto, Takeshi Kitai, Yugo Yamashita, Moritake Iguchi, Kazuya Nagao, Yuichi Kawase, Takashi Morinaga, Mamoru Toyofuku, Yutaka Furukawa, Kenji Ando, Kazushige Kadota, Yukihito Sato, Koichiro Kuwahara.

**Methodology:** Yuta Seko, Takao Kato, Takeshi Morimoto, Hidenori Yaku, Yasutaka Inuzuka, Yodo Tamaki, Neiko Ozasa, Masayuki Shiba, Erika Yamamoto, Yusuke Yoshikawa, Takeshi Kitai, Yugo Yamashita, Moritake Iguchi, Kazuya Nagao, Yuichi Kawase, Takashi Morinaga, Mamoru Toyofuku, Yutaka Furukawa, Kenji Ando, Kazushige Kadota, Yukihito Sato, Koichiro Kuwahara, Takeshi Kimura.

**Project administration:** Takao Kato, Hidenori Yaku, Yasutaka Inuzuka, Yodo Tamaki, Neiko Ozasa, Erika Yamamoto, Takeshi Kimura.

**Supervision:** Takeshi Kimura.

**Writing – original draft:** Yuta Seko.

**Writing – review & editing:** Takao Kato, Takeshi Morimoto, Hidenori Yaku, Yasutaka Inuzuka, Yodo Tamaki, Neiko Ozasa, Masayuki Shiba, Erika Yamamoto, Takeshi Kitai, Yugo Yamashita, Moritake Iguchi, Kazuya Nagao, Yuichi Kawase, Takashi Morinaga, Mamoru Toyofuku, Yutaka Furukawa, Kenji Ando, Kazushige Kadota, Yukihito Sato, Koichiro Kuwahara, Takeshi Kimura.

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
