## [Decision Letter · Decision Letter 0]

18 Apr 2023

PONE-D-22-33948Weight Loss During Follow-up in Patients with Acute Heart Failure: from the KCHF registryPLOS ONE

Dear Dr. Kato,

Thank you for submitting your manuscript to PLOS ONE. After careful consideration, we feel that it has merit but does not fully meet PLOS ONE’s publication criteria as it currently stands. Therefore, we invite you to submit a revised version of the manuscript that addresses the points raised during the review process.

We look forward to receiving your revised manuscript.

Kind regards,

Yoshihiro Fukumoto

Academic Editor

PLOS ONE

2. Thank you for submitting the above manuscript to PLOS ONE. During our internal evaluation of the manuscript, we found significant text overlap between your submission and previous work in the Discussion and Limitations.

Please revise the manuscript to rephrase the duplicated text, cite your sources, and provide details as to how the current manuscript advances on previous work. Please note that further consideration is dependent on the submission of a manuscript that addresses these concerns about the overlap in text with published work.

We will carefully review your manuscript upon resubmission and further consideration of the manuscript is dependent on the text overlap being addressed in full. Please ensure that your revision is thorough as failure to address the concerns to our satisfaction may result in your submission not being considered further.

“This study was supported by grant 18059186 from the Japan Agency for Medical Research and Development (Drs T. Kato, Kuwahara, and Ozasa).”

4. Thank you for stating the following in the Acknowledgments/ Funding Section of your manuscript:

“This study was supported by grant 18059186 from the Japan Agency for Medical Research and Development (Drs T. Kato, Kuwahara, and Ozasa).”

“This study was supported by grant 18059186 from the Japan Agency for Medical Research and Development (Drs T. Kato, Kuwahara, and Ozasa).”

Reviewers' comments:

Reviewer's Responses to Questions

**Comments to the Author**

1. Is the manuscript technically sound, and do the data support the conclusions?

Reviewer #1: Yes

Reviewer #2: Yes

2. Has the statistical analysis been performed appropriately and rigorously? 

Reviewer #1: Yes

Reviewer #2: Yes

3. Have the authors made all data underlying the findings in their manuscript fully available?

Reviewer #1: No

Reviewer #2: Yes

4. Is the manuscript presented in an intelligible fashion and written in standard English?

Reviewer #1: Yes

Reviewer #2: Yes

5. Review Comments to the Author

Reviewer #1: In this prospective observational study, the authors analyzed the correlates and prognostic implications of weight loss at 6-month follow-up in patients discharged after an episode of AHF. Interestingly, weight loss was associated with worse clinical phenotype and outcomes.

The authors should be congratulated for the original idea and the design of the study.

Some points should be addressed to improve the quality of the work:

- How was the 5% cut-off selected? Please detail.

- Beyond the pre-specified cut-off, was weight loss linearly related to worse outcome? A spline curve could express this graphically.

- No difference was reported in NT-proBNP levels at follow-up visit, despite the well-known correlation with body composition (e.g., 10.1016/j.jchf.2021.05.014). Please discuss.

- It has been reported that the interaction between body composition and outcomes in HF patients may depend on HF etiology (i.e., ischemic vs. non-ischemic, 10.1177/2047487320927610). Please discuss.

- The lack of physical training is a well-known determinant of both weight loss and poor outcome in HF patients. Did any patient undergo cardiac rehabilitation or training programs after the index hospitalization? In any case this should be matter of discussion.

Reviewer #2: The manuscript by Seko and colleagues entitled “Weight Loss During Follow-up in Patients with Acute Heart Failure: from the KCHF registry” is very interesting.

The management of weight in patients with heart failure is a huge problem. In general, weight gain suggests worsening of heart failure, and physicians and patients often aim to maintain or reduce body weight. On the other hand, this paper is highly significant in that it shows that weight loss does not necessarily lead to a favorable prognosis, as pointed out in this paper.

However, I have some comments.

Major comments

1.In patients with heart failure, weight loss is not all bad, and there appears to be malignant weight loss as shown in this paper and benign weight loss in patients who need “intended” reduction of body weight. Therefore, I think, if possible, it would be better to add subgroup analyzes such as baseline body weight-specific analysis.

2.To prevent weight loss in patients with heart failure, nutrition management and maintenance of activity are important, as the authors also stated. If possible, a more detailed baseline assessment of nutritional status (such as geriatric nutritional risk index) should be added. Among them, if the clinical characteristics of the group that loses weight in a group with maintained nutritional status is clarified, the target of intervention will be clarified a little more.

3.Why is the KM curve for all-cause death at 30 days getting different? I think factors other than weight loss are important.

Minor comment

It may be problematic that about 17% of all patients were analyzed.

6. PLOS authors have the option to publish the peer review history of their article (what does this mean?). If published, this will include your full peer review and any attached files.

Reviewer #1: No

Reviewer #2: **Yes: **Hiroaki Kitaoka

---

## [Author Response · Author response to Decision Letter 0]

20 May 2023

Response

We thank the reviewers for careful assessment and positive comments. The manuscript has been rechecked and the necessary changes have been made in accordance with the editors’ and reviewers’ suggestions.

 

Reviewer #1: In this prospective observational study, the authors analyzed the correlates and prognostic implications of weight loss at 6-month follow-up in patients discharged after an episode of AHF. Interestingly, weight loss was associated with worse clinical phenotype and outcomes.

The authors should be congratulated for the original idea and the design of the study.

Some points should be addressed to improve the quality of the work:

- How was the 5% cut-off selected? Please detail.

Response

Thank you for your valuable comment. We have selected 5% cut-off because cachexia is primarily defined as a loss of body mass of more than 5% in 12 months and previous studies also have selected 5% cut-off. (Reference [4, 6]) We have included some references in method section. (Page 8, line 12)

- Beyond the pre-specified cut-off, was weight loss linearly related to worse outcome? A spline curve could express this graphically.

Response

Thank you for your careful assessment and suggestions. A spline curve could express weight loss linearly related to worse outcome graphically, but it is difficult to be used in this study in terms of low patient numbers. We have performed additionally sensitivity analysis classified into three groups as follows: ≥10% decrease in body weight, -10%< body weight change ≤ -5% and no weight loss (reference, -5%< body weight change) (S2 Fig). Higher decrease in body weight was related to worse outcome.

- No difference was reported in NT-proBNP levels at follow-up visit, despite the well-known correlation with body composition (e.g., 10.1016/j.jchf.2021.05.014). Please discuss.

Response

Thank you for your careful assessment. We have included above reference and discussion as follows: “In our study, no difference was reported in NT-proBNP levels at follow-up visit due to the small number of patients with data and large variability.”(Page 19, line 10-12)

- It has been reported that the interaction between body composition and outcomes in HF patients may depend on HF etiology (i.e., ischemic vs. non-ischemic, 10.1177/2047487320927610). Please discuss.

Response

Thank you for your careful assessment. Gentile et al. reported that higher BMI were associated with better outcome in non-ischaemic heart failure, but not in ischaemic

heart failure. (Eur J Prev Cardiol. 2021;28(9):948-55.)

In our study, we focus on weight loss, but we have evaluated the interactions between the ischemic etiology and the effects of weight loss relative to no weight loss on all-cause death.

The ratio of ischemic etiology was not significant difference between weight loss and no weight loss group (Table 1). In addition, there was not interaction between weight loss and outcomes may depend on HF etiology in subgroup analysis (S5 Fig).

- The lack of physical training is a well-known determinant of both weight loss and poor outcome in HF patients. Did any patient undergo cardiac rehabilitation or training programs after the index hospitalization? In any case this should be matter of discussion.

Response

Thank you for your careful assessment. In this study, we have no data about undergoing cardiac rehabilitation or training programs after the index hospitalization. We have added the next sentence in limitation section. “The lack of physical training is a well-known determinant of both weight loss and poor outcome in HF patients, but we have no data about undergoing cardiac rehabilitation or training programs after the index hospitalization.” (Page 20, line 11-13)

Reviewer #2: The manuscript by Seko and colleagues entitled “Weight Loss During Follow-up in Patients with Acute Heart Failure: from the KCHF registry” is very interesting.

The management of weight in patients with heart failure is a huge problem. In general, weight gain suggests worsening of heart failure, and physicians and patients often aim to maintain or reduce body weight. On the other hand, this paper is highly significant in that it shows that weight loss does not necessarily lead to a favorable prognosis, as pointed out in this paper.

However, I have some comments.

Major comments

1.In patients with heart failure, weight loss is not all bad, and there appears to be malignant weight loss as shown in this paper and benign weight loss in patients who need “intended” reduction of body weight. Therefore, I think, if possible, it would be better to add subgroup analyzes such as baseline body weight-specific analysis.

Response

Thank you for your careful assessment and valuable comments. We have evaluated the interactions between the subgroup factors such as age, etiology of HF, LVEF, BMI, albumin, anemia, geriatric nutritional risk index (GNRI) and the effects of weight loss relative to no weight loss on all-cause death after 6-month visit. There was no significant interaction between the effect of weight loss relative to no weight loss on the primary outcome measure and the subgroup factors (S5 Fig).

We have includes the next sentences as follows: “In addition, there was a trend that patients with low BMI, low albumin levels, or low GNRI at 6-month visit were lily to have more risk when compared to those without, although there was no significant interaction between the effect of weight loss relative to no weight loss on the primary outcome measure and the subgroup factors.” (Page 19, line 18-22)

2.To prevent weight loss in patients with heart failure, nutrition management and maintenance of activity are important, as the authors also stated. If possible, a more detailed baseline assessment of nutritional status (such as geriatric nutritional risk index) should be added. Among them, if the clinical characteristics of the group that loses weight in a group with maintained nutritional status is clarified, the target of intervention will be clarified a little more.

Response

Thank you for your careful assessment and valuable comments. We have added the data of geriatric nutritional risk index and its change in Table 1. GNRI was lower and the increase of GNRI was less in the weight loss group (Table 1). We have included the next sentences as follows: “The increase of GNRI was less in the BW loss group. These data suggest the importance of nutrition management and maintenance of activity during the follow-up for patients with AHF.” (Page 19, line 16-18)

3.Why is the KM curve for all-cause death at 30 days getting different? I think factors other than weight loss are important.

Response

Thank you for your careful assessment and valuable comments. It is not clear why the KM curves for all-cause mortality at 30 days differ. There are two suggested reasons for this: 1) the number of patients enrolled in this study is relatively small and the KM curves for the two cohorts are not smoothed; 2) it is possible that some of the patients who lost weight after discharge may have further improved congestion, making it difficult for them to have events immediately after the starting point of 6 months after discharge. We have added into the Discussion sections (Page 18, line 11-15)

Minor comment

It may be problematic that about 17% of all patients were analyzed.

Response

Thank you for your careful assessment. We agree with your opinion and described it in the Limitation.

---

## [Decision Letter · Decision Letter 1]

12 Jun 2023

Weight Loss During Follow-up in Patients with Acute Heart Failure: from the KCHF registry

PONE-D-22-33948R1

Dear Dr. Kato,

We’re pleased to inform you that your manuscript has been judged scientifically suitable for publication and will be formally accepted for publication once it meets all outstanding technical requirements.

Kind regards,

Yoshihiro Fukumoto

Academic Editor

PLOS ONE

Additional Editor Comments (optional):

Reviewers' comments:

Reviewer's Responses to Questions

**Comments to the Author**

1. If the authors have adequately addressed your comments raised in a previous round of review and you feel that this manuscript is now acceptable for publication, you may indicate that here to bypass the “Comments to the Author” section, enter your conflict of interest statement in the “Confidential to Editor” section, and submit your "Accept" recommendation.

Reviewer #1: (No Response)

Reviewer #2: All comments have been addressed

2. Is the manuscript technically sound, and do the data support the conclusions?

Reviewer #1: (No Response)

Reviewer #2: Yes

3. Has the statistical analysis been performed appropriately and rigorously? 

Reviewer #1: (No Response)

Reviewer #2: Yes

4. Have the authors made all data underlying the findings in their manuscript fully available?

Reviewer #1: (No Response)

Reviewer #2: Yes

5. Is the manuscript presented in an intelligible fashion and written in standard English?

Reviewer #1: (No Response)

Reviewer #2: Yes

6. Review Comments to the Author

Reviewer #1: The authrors have signficantly improved the overall quality of their manuscript. I have no further comment.

Reviewer #2: The previously mentioned issues have been adequately revised. I think it will have a impact on clinical practice.

7. PLOS authors have the option to publish the peer review history of their article (what does this mean?). If published, this will include your full peer review and any attached files.

Reviewer #1: No

Reviewer #2: No

---

## [Editor Report · Acceptance letter]

15 Jun 2023

PONE-D-22-33948R1 

Weight Loss During Follow-up in Patients with Acute Heart Failure: from the KCHF registry 

Dear Dr. Kato:

I'm pleased to inform you that your manuscript has been deemed suitable for publication in PLOS ONE. Congratulations! Your manuscript is now with our production department. 

Kind regards, 

on behalf of

Dr. Yoshihiro Fukumoto 

Academic Editor

PLOS ONE